# Ion diffusion retarded by diverging chemical susceptibility

Yuhang Cai [1,2,7], Zhaowu Wang [3,4,7], Jiawei Wan [1,2], Jiachen Li[1,2], Ruihan Guo[1,2], Joel W. Ager [1,2], Ali Javey [2,5], Haimei Zheng [1,2], Jun Jiang [6] & Junqiao Wu [1,2] ✉

For first-order phase transitions, the second derivatives of Gibbs free energy (specific heat and compressibility) diverge at the transition point, resulting in an effect known as super-elasticity along the pressure axis, or super-thermicity along the temperature axis. Here we report a chemical analogy of these singularity effects along the atomic doping axis, where the second derivative of Gibbs free energy (chemical susceptibility) diverges at the transition point, leading to an anomalously high energy barrier for dopant diffusion in co-existing phases, an effect we coin as super-susceptibility. The effect is realized in hydrogen diffusion in vanadium dioxide ($VO_2$) with a metal-insulator transition (MIT). We show that hydrogen faces three times higher energy barrier and over one order of magnitude lower diffusivity when it diffuses across a metal-insulator domain wall in $VO_2$. The additional energy barrier is attributed to a volumetric energy penalty that the diffusers need to pay for the reduction of latent heat. The super-susceptibility and resultant retarded atomic diffusion are expected to exist universally in all phase transformations where the transformation temperature is coupled to chemical composition, and inspires new ways to engineer dopant diffusion in phase-coexisting material systems.

First order phase transitions feature a discontinuous first derivative of Gibbs free energy ($G$) at the transition point[1]. Depending on the driven force being temperature ($T$) or pressure ($P$) [or in the uniaxial case, stress ($\sigma$)], the first derivative corresponds to entropy $S = -\left(\frac{\partial G}{\partial T}\right)_P$ or volume $V = \left(\frac{\partial G}{\partial P}\right)_T$ [or uniaxial strain $\varepsilon = -\left(\frac{\partial G}{\partial \sigma}\right)_T$]. The second-order derivative of $G$ diverges at the transition point, corresponding to specific heat $C_P = T\left(\frac{\partial S}{\partial T}\right)_P$ or compressibility $\beta = -\frac{1}{V}\left(\frac{\partial V}{\partial P}\right)_T$ [or uniaxial compressibility $\beta = -\left(\frac{\partial \varepsilon}{\partial \sigma}\right)_T$], as shown schematically in Fig. 1a. In systems when the two phases co-exist, the diverging second derivative of $G$ is manifested as an infinitely low stiffness or infinitely high heat absorptivity[2,3], an effect known as super-elasticity (along the $P$ or $\sigma$ axis) or super-thermicity (along the $T$ axis). For example, when water is in equilibrium with vapor at the boiling point, the entire system

absorbs heat without raising its temperature (divergent specific heat), or shrinks volume without raising pressure (divergent compressibility). In addition to $T$ and $\varepsilon$, dynamic variation of chemical composition through dopant diffusion may drive the phase transition[4], as $G$ can be affected by chemical doping[5]. However, it has remained a technical challenge to control and quantify atomic diffusion near the phase transition[4,6-9]. As a result, unlike the effects of super-elasticity and super-thermicity, the diverging second derivative of $G$ along the doping concentration ($x$) axis and its effects on physical properties have rarely been explored.

In this work, we investigate the chemical analogy of super-elasticity using the first-order, metal-insulator transition in vanadium dioxide as a model system. We show that a similar effect, termed as

[1]Department of Materials Science and Engineering, University of California, Berkeley, Berkeley, CA 94720, USA. [2]Materials Sciences Division, Lawrence Berkeley National Laboratory, Berkeley, CA 94720, USA. [3]School of Science, Hebei University of Technology, Tianjin 300401, China. [4]National Laboratory of Solid State Microstructures, Nanjing University, Nanjing 210093, China. [5]Department of Electrical Engineering and Computer Science, University of California, Berkeley, Berkeley, CA 94720, USA. [6]Key Laboratory of Precision and Intelligent Chemistry, School of Chemistry and Materials Science, University of Science and Technology of China, Hefei 230026 Anhui, China. [7]These authors contributed equally: Yuhang Cai, Zhaowu Wang ✉e-mail: wuj@berkeley.edu

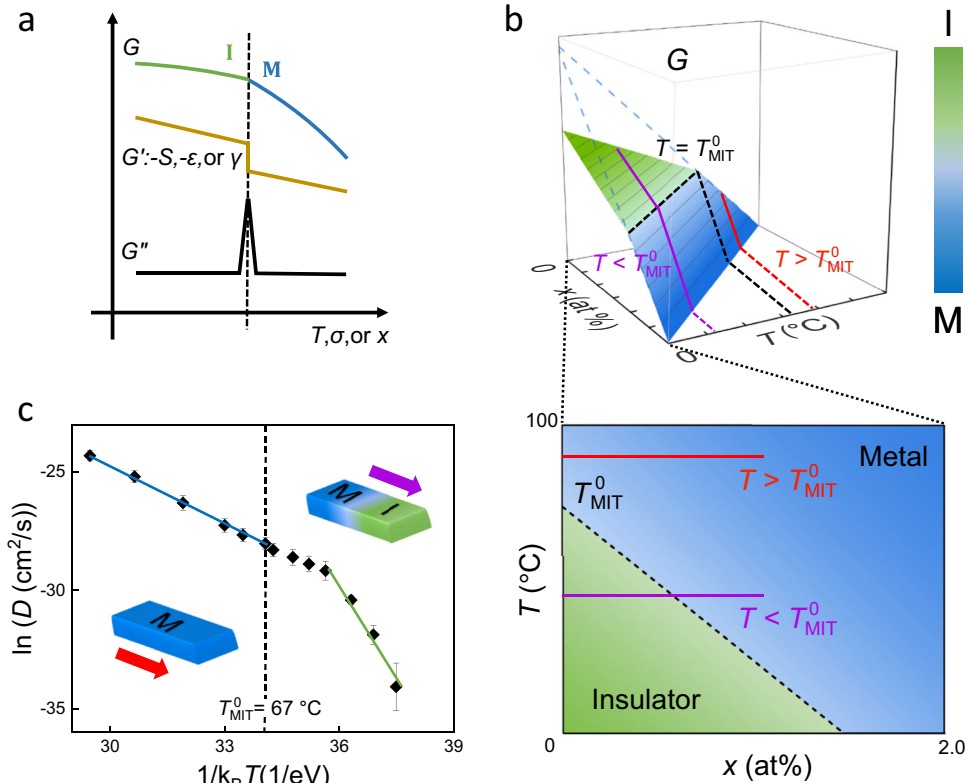

**Fig. 1 | Super-susceptibility and its analogy to super-elasticity. a** Gibbs free energy ($G$) as a function of temperature ($T$), stress ($\sigma$), or doping fraction ($x$) at the phase transition point. Also shown is its first derivative ($G$), giving to discontinuous entropy ($S = -\left(\frac{\partial G}{\partial T}\right)_{\sigma,x}$), strain ($\varepsilon = -\left(\frac{\partial G}{\partial \sigma}\right)_{T,x}$) and formation energy ($\gamma = \left(\frac{\partial G}{\partial x}\right)_{T,\sigma}$), as well as its divergent second derivative ($G$", proportional to specific heat $C = T\left(\frac{\partial S}{\partial T}\right)_{\sigma,x}$, compressibility $\beta = \left(\frac{\partial \varepsilon}{\partial \sigma}\right)_{T,x}$, and chemical susceptibility $\alpha = \left(\frac{\partial \gamma}{\partial x}\right)_{T,\sigma}$). **b** Gibbs free energy of the metal (M) and insulator (I) phases of $VO_2$ as a function of

temperature ($T$) and hydrogen doping fraction ($x$). Lower panel shows the phase diagram in the $T$ - $x$ plane. The two solid lines (red and purple) show the distinct paths for doping at $T$ higher or lower than $T_{MIT}^0$, respectively. **c** Observed effective diffusivity of hydrogen in $VO_2$, showing a normal, Arrhenius dependence on $T$ in the pure M phase, but an abnormal retardation in the M-I phase coexisting regime at lower temperatures. Blue and green solid lines are drawn to guide the eye. Error bar: uncertainty of ln$D$ (see Supplementary Note 2).

super-susceptibility, occurs in hydrogen doping of $VO_2$, resulting in an order of magnitude retardation in effective diffusivity of hydrogen in $VO_2$ when its metal (M) and insulator (I) phases co-exist. As the mechanism is fundamentally based on the thermodynamics of the phase transition, the effect is expected to exist universally, albeit to different extents, in most first-order phase transitions. From a practical perspective, the effect would impact ion diffusion dynamics and kinetics in material systems with coexisting two phases that are linked via a first order phase transition, such as hydrogen storage and rechargeable ion battery.

## Results

### Energy analysis of hydrogen doping in VO₂

Strain-free $VO_2$ single crystals undergo a sharp MIT at the temperature $T_{MIT}^0 = 67\,°C$, accompanied by a structural transition from the monoclinic, I phase at lower temperatures to the rutile, M phase at higher temperatures. The entropy change at the thermally driven MIT is $\Delta S = \Delta H/T_{MIT}^0$, where $\Delta H = 1020$ Cal/mol is enthalpy change (latent heat) at $T = T_{MIT}^0$. Interestingly, the MIT can also be driven mechanically with a uniaxial compression along the [100] direction in the I phase at room temperature[10,11], resulting in a length shrinkage with a spontaneous strain of $\Delta \varepsilon = 1\%$, which is also related to the enthalpy change via $\Delta \varepsilon = -\Delta(\Delta H)/\sigma_{MIT}^0$ [12]. Here $\sigma_{MIT}^0$ is the critical compressive stress of ~4 GPa driving the MIT[13,14], and $\Delta(\Delta H)$ is the reduction in latent heat at room temperature from that at $T_{MIT}^0$.

It is also known that the MIT can be chemically driven by doping with transition metals or light ions such as hydrogen, the latter rendering $VO_2$ a potential hydrogen storage medium[7,15]. Specifically for hydrogen

doping, using catalysts like Platinum (Pt) or Palladium (Pd) to crack hydrogen molecules into atoms[16], it has been shown that ~0.9 at% hydrogen doping drives an I-phase $VO_2$ into M phase at room temperature[17,18]. Moreover, first-principles calculations have shown that hydrogen doping reduces the Gibbs free energy of both phases[19], with the reduction rate equal to the formation energy ($\gamma$) of the doped system. This process is analogous to the thermal and mechanical driving of the MIT, allowing us to write the differential Gibbs free energy as the sum of all (Supplementary Note 1 and Supplementary Table 1),

$$dG = -SdT - \varepsilon d\sigma + \gamma dx. \tag{1}$$

Figure 1b schematically plots $G$ of $VO_2$ as a function of $T$ and hydrogen atomic fraction ($x$), showing the two surfaces of I phase and M phase intersecting at a line where the MIT occurs. The lower panel of Fig. 1b shows the projection of the two surfaces in the $T$ - $x$ plane, defining the phase diagram of $VO_2$ in the plane which was also experimentally confirmed[20]. It can be seen that the temperature of MIT, $T_{MIT}$, depends critically on $x$. This is attributed to higher occupancy of the vanadium $3d$ band by free electrons donated from the hydrogen dopants, which tends to collapse the bandgap and lower $T_{MIT}$[4]. First-principles calculations have provided a quantitative $T_{MIT}$ - $x$ relationship[19]. The Clausius-Clapeyron equation describes the slope of dependence as

$$\frac{dT_{MIT}}{dx} = \frac{\Delta\gamma \cdot T_{MIT}^0}{\Delta H}, \tag{2}$$

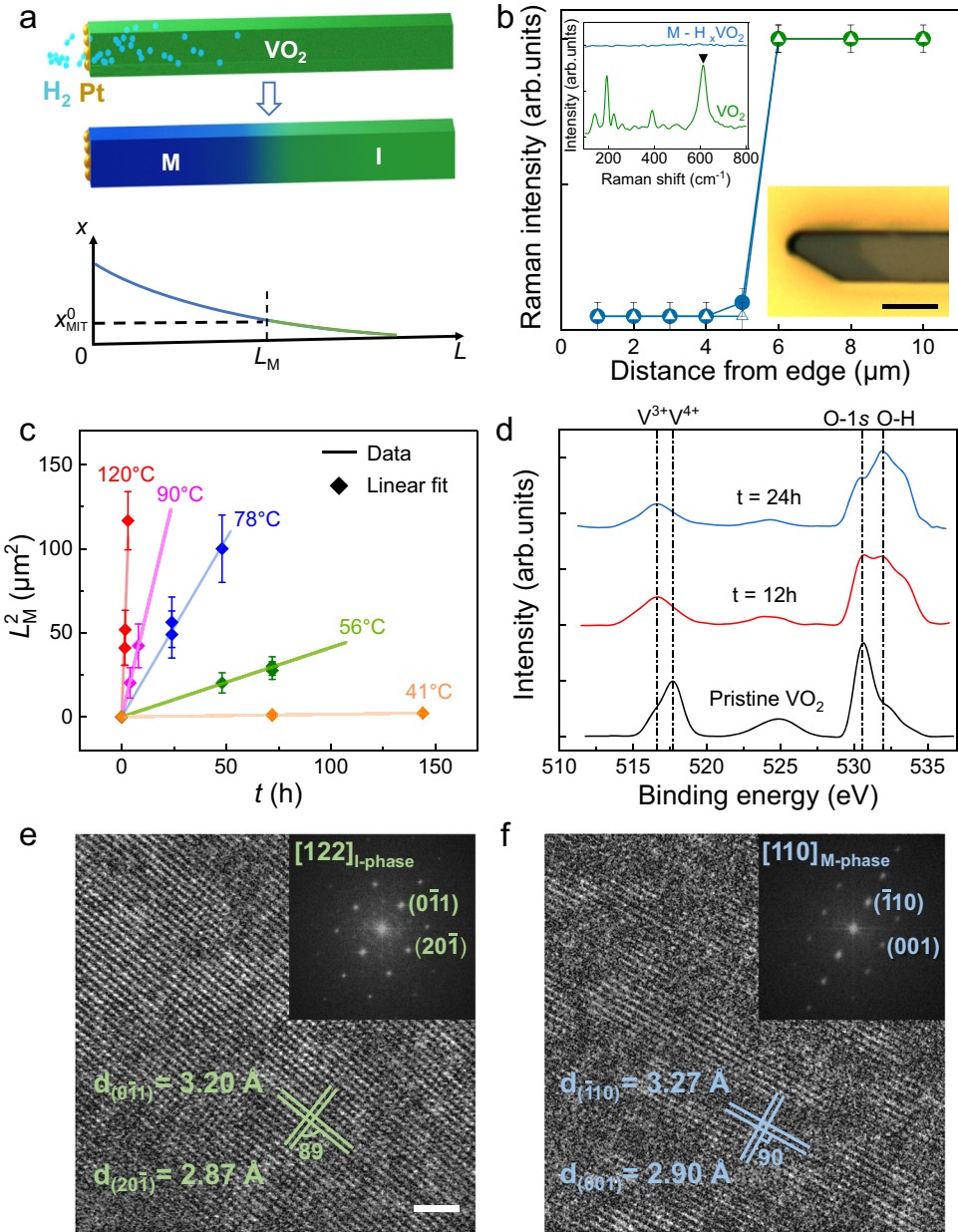

**Fig. 2 | Characterization of diffusion. a** Schematic of a VO$_2$ microbeam subject to one-dimensional hydrogen diffusion from the Pt-catalyzed tip at $T_{anneal}$. The M-I domain wall observed at room temperature determines the M-domain length $L_M$. $L$ represents the **b**eam direction. **b** Raman intensity of the main peak at 612 cm$^{-1}$ as a function of distance from the catalyzed microbeam tip for two samples (hollow triangles and solid circles) both annealed at 56 °C for 72 h. Error bar: reading error of Raman peak intensity. Inset shows representative Raman spectra of M and I phases, and optical image of a microbeam showing the M-I domain wall. Scale bar:

5 µm. **c** $L_M^2$ plotted as a function of diffusion time $t$. The slope defines the effective diffusivity. Error bar: uncertainty of measured $L_M^2$. **d** XPS spectra of pristine VO$_2$ and of VO$_2$ with hydrogen diffusion at 120 °C for 12 h and 24 h, respectively, showing the formation of more O-H bonds with longer time hydrogenation. **e** High-resolution TEM image of pristine VO$_2$ indexed to the I-phase (monoclinic structure), and **f** of hydrogen diffused VO$_2$ indexed to the M-phase (rutile structure) with a slight lattice expansion, attributed to the interstitial hydrogen doping. Scale bar: 2 nm.

where $T_{MIT}^0 = 67$ °C is $T_{MIT}$ at zero $x$, and $\Delta\gamma = \gamma_M - \gamma_I$ is the change in formation energy across the MIT. The Clausius-Clapeyron equation is fundamentally related to the fact that chemical doping reduces the latent heat at a rate that is proportional to the doping fraction, $T_{MIT}\Delta S - T_{MIT}^0\Delta S = x\Delta\gamma$. In a linear approximation, we define the threshold $x$ at room temperature ($T_{RT} = 20$ °C) as $x_{MIT}^0 = 0.9\%$[19], the phase boundary of $T_{MIT}$ versus $x$ is written as $(T_{MIT} - T_{MIT}^0)/(T_{RT} - T_{MIT}^0) = x/x_{MIT}^0$.

The negative slope of the phase boundary provides a useful means to reduce $T_{MIT}$ with hydrogen doping for practical applications[21–23]. It also offers a way to experimentally investigate the

chemical analogy of the super-elasticity effect along the $x$ axis. The $T$-$\sigma$ phase diagram along the uniaxial $[1,0,0]_{I-phase}$ (or equivalently $[0,0,1]_{M-phase}$ direction) of VO$_2$ (Supplementary Fig. 1) has a similar negative-sloped phase boundary as the $T$-$x$ phase diagram (Fig. 1b). If a uniaxial compression is applied at a temperature $T > T_{MIT}^0$, the system would respond normally following a linear $\varepsilon$-$\sigma$ curve with a slope equal to the isothermal compressibility ($\beta = d\varepsilon/d\sigma = 1/Y$, where $Y$ is Young's modulus). However, if the uniaxial stress is applied at a temperature $T < T_{MIT}^0$, the system would exhibit a jump in the $\varepsilon$-$\sigma$ relation where the compressibility diverges. This is because when $\sigma$ raises to the critical stress $\sigma_{MIT}$, the system is mechanically driven through the MIT from

the I phase into the M phase[10], abruptly shrinking in length with an ferroelastic strain of $\Delta\varepsilon \approx 1\%$. Such super-elasticity exists in a wide range of phase transitions such as the austenite/martensite phase transformation[24].

Now, in the $T$ - $x$ phase diagram as shown in Fig. 1b, when isothermally diffusing hydrogen into VO$_2$ at $T > T_{MIT}^0$, the system is always in the M phase, hence a normal diffusion process is anticipated with a constant formation energy $\gamma = \gamma_M$. When diffusing hydrogen into I-phase VO$_2$ at $T < T_{MIT}^0$, however, the diffused hydrogen will first convert the surface into a M phase layer owing to the reduction of $T_{MIT}$ by hydrogen doping. Further diffusion of hydrogen will have to cross the M-I phase boundary, experiencing a discontinuous rise in formation energy from $\gamma_M$ to $\gamma_I$ by $-\Delta\gamma$. As a result, the chemical susceptibility ($\alpha = d\gamma/dx$) diverges. The change in latent heat can be written as $\Delta(\Delta H) = x_{MIT}\Delta\gamma$, akin to the equation when $T$ or $\sigma$ is the variable, $\Delta(\Delta H) = \Delta T_{MIT}\Delta S$ or $-\sigma_{MIT}\Delta\varepsilon$. When $T$ or $\sigma$ is the variable, the energy needed to pay across the MIT is provided by the heat absorbed ($T_{MIT}\Delta S$) or the work done ($\sigma_{MIT}\Delta\varepsilon$), respectively. In the case of hydrogen doping, this is provided by the chemical energy ($|x_{MIT}\Delta\gamma|$) contributed from the diffusing ions. Therefore, one expects that the diffusion energetics will be different at $T < T_{MIT}^0$ from the normal, single-phase case of $T > T_{MIT}^0$.

## Retarded diffusivity of hydrogen in VO$_2$

We have measured the effective diffusivity ($D$) of hydrogen in VO$_2$ at different temperatures (Fig. 1c and Supplementary Note 2). Our results show that indeed, $D$ shows a normal, Arrhenius dependence on temperature at $T > T_{MIT}^0$; but at $T < T_{MIT}^0$, a large deviation is seen from the extrapolation of the high-temperature Arrhenius dependence, showing ~27 times lower diffusivity at 37 °C, the lowest temperature at which measurement was carried out, corresponding to a three-fold increase in diffusion energy barrier.

The VO$_2$ samples were grown as single-crystal microbeams with axis along the $[0,0,1]_{M\text{-phase}}$ (or $[1,0,0]_{I\text{-phase}}$) direction using the Vapor–Liquid–Solid method[25] and dry transferred onto new substrates to release strain[9]. The high quality of these strain-free microbeams is supported by a sharp MIT observed at the expected 67 °C (Supplementary Fig. 3). To initiate hydrogen diffusion in the relatively low temperature range, nano-sized Pt catalysts (Supplementary Fig. 2) were deposited onto only the tip area of the microbeams which was exposed using photolithography (details in Supplementary Fig. 3). Subsequently, the Pt/VO$_2$ samples were annealed at temperature $T_{anneal}$ under forming gas containing 10% hydrogen molecules, so that the split hydrogen atoms diffused in from the tip of the microbeam primarily along the $[0,0,1]_{M\text{-phase}}$ axis (Fig. 2a). Such a quasi-one-dimensional (1D) diffusion resulted in a graded doping of hydrogen along the microbeam. Returning to room temperature, the high-$x$ side of the microbeam is in the M phase because its $x$ is already higher than $x_{MIT}^0$. The distinct optical contrast of M and I phases of VO$_2$ allows us to visually discern the M-I domain wall at room temperature to determine the length of the M domain, $L_M$ (Fig. 2b). When the microbeams were heated toward $T_{MIT}^0$, the M-I domain wall gradually shifted toward the other tip until the entire microbeam turns into M phase (Supplementary Fig. 4), consistent with the 1D diffusion and resultant graded hydrogen doping along the axis of the microbeam[26]. Control experiments were also designed and performed which revealed that the diffusion perpendicular to the microbeam axial direction is four orders of magnitude slower than along the axial direction (Supplementary Fig. 5c, d).

For more accurate determination of $L_M$, Raman spectroscopy was used. The Inset in Fig. 2b shows room-temperature Raman spectrum of monoclinic I-phase of VO$_2$ beyond the M-I wall of the microbeam, and a featureless Raman spectrum within the hydrogenated segment consistent with the M phase. This offers a reliable method to resolve the M-I domain wall and $L_M$ within the diffraction limit (<1 μm) by mapping

the intensity of the Raman peak at ~612 cm$^{-1}$ (Fig. 2b and Supplementary Fig. 5a, b. A low laser power (0.01 mW) was used in the Raman measurements to minimize potential heating from the laser beam. Figure 2d–f further confirm the hydrogenation and resultant conversion from I phase to M phase by x-ray photoelectron spectroscopy (XPS)[27] and high-resolution transmission electron microscopy (HRTEM)[28], more detailed structural information and morphology of synthesized microbeams is shown and discussed in Supplementary Figs. 7−9 and Supplementary Note 3. While the lattice spacings in Fig. 2e correspond to those of ($20\bar{1}$) and ($0\bar{1}1$) planes of I-phase, monoclinic VO$_2$ (Supplementary Fig. 9c, d)[29], the HRTEM image of hydrogenated VO$_2$ in Fig. 2f can be indexed as a M-phase, rutile structure with a slightly expanded lattice (Supplementary Table 3), attributed to interstitial hydrogen doping[7]. In addition, electrical measurements were conducted on both the pristine and hydrogenated VO$_2$ to verify the formation of M phase by atomic hydrogen diffusion (Supplementary Fig. 6 and Supplementary Table 2).

For 1D diffusion in a semi-infinite system with a fixed concentration at the surface, the solution to the Fick's second law is given by the error function. Following the definition of $L_M$ as the length from the diffusion inlet (the microbeam tip) to the position where $x$ reaches a fixed value of $x_{MIT}^0$, it can be shown that $L_M$ is numerically proportional to the diffusion length $\sqrt{Dt}$. In Fig. 2c we plot the measured $L_M^2$ as a function of the diffusion time $t$ at a range of temperatures. The linear dependence in Fig. 2c further validates the analysis based on the 1D diffusion model. The slope of the dependence is taken as the effective diffusivity, $D(T)$, within the difference of a $T$-independent numerical factor. According to the thermal activation mechanism of atomic diffusion, the temperature dependence of diffusivity can be written as $D(T) = D_0 \exp\left(-\frac{E}{k_B T}\right)$, where $D_0$ is a constant and $E$ is the diffusion energy barrier. $D(T)$ is plotted in Fig. 1c, which shows a good Arrhenius dependence in the pure M phase ($T = T_{anneal} > T_{MIT}^0$) with a fitted diffusion energy barrier $E = E_M^{exp} = 0.8 \pm 0.1$ eV, comparable to reported theoretical or experimental activation energy for hydrogen diffusion in tetragonal VO$_2$[6,9] and TiO$_2$[8,30]. However, $D(T)$ deviates significantly from the same Arrhenius dependence at lower temperatures ($T_{anneal} < T_{MIT}^0$). For example, at $T_{anneal} = 37$ °C, the measured $D$ is ~27 times lower than the $D$ value extrapolated from the $T_{anneal} > T_{MIT}^0$ region. It can be concluded that, therefore, hydrogen diffusion in the I phase faces a much higher energy barrier than in the M phase. A forced Arrhenius fit to $D(T)$ in the low-temperature regime would yield an energy barrier $E = E_I^{exp} = 2.2$ eV, a nearly three-fold increase from $E_M^{exp}$.

## First-principles calculation and extra diffusion energy barrier

It is known that the M and I phases have rutile and monoclinic crystal structure, respectively, as confirmed by TEM in Fig. 2e, f. As a result, the diffusion energy barrier $E$ might take different values in these two structures. We carried out first-principles calculations to evaluate the energy barriers for hydrogen ions to diffuse in the two phases. Supercells of monoclinic (I-phase) and rutile (M-phase) VO$_2$ containing the same hydrogen concentration (one hydrogen atom per V$_4$O$_8$ supercell) were built, with the diffusion directions of $[1,0,0]_{I\text{-phase}}$ and $[0,0,1]_{M\text{-phase}}$ considered, respectively (Fig. 3a, b). In Fig. 3c, d, the minimum energy paths are shown which represent the trajectory wherein the energy disparity between the highest and lowest points is minimal (Supplementary Fig. 10). For the M-phase, a much greater energy barrier along the $[1,1,0]$ direction (1.5 eV) than along the $[0,0,1]$ direction (0.6 eV) is seen (Fig. 3d, e), which is consistent with our observation that the diffusion along the microbeam axis ($[0,0,1]$) is much faster than perpendicular to the axis direction in the same, pure M phase.

More importantly, the calculated diffusion energy barrier along the microbeam axis direction does not vary drastically between the two pure phases: it is calculated to be $E_I^{DFT} = 0.9$ eV in the I-phase along $[1,0,0]$, while it is $E_M^{DFT} = 0.6$ eV in the M-phase along $[0,0,1]$ (Fig. 3c, d).

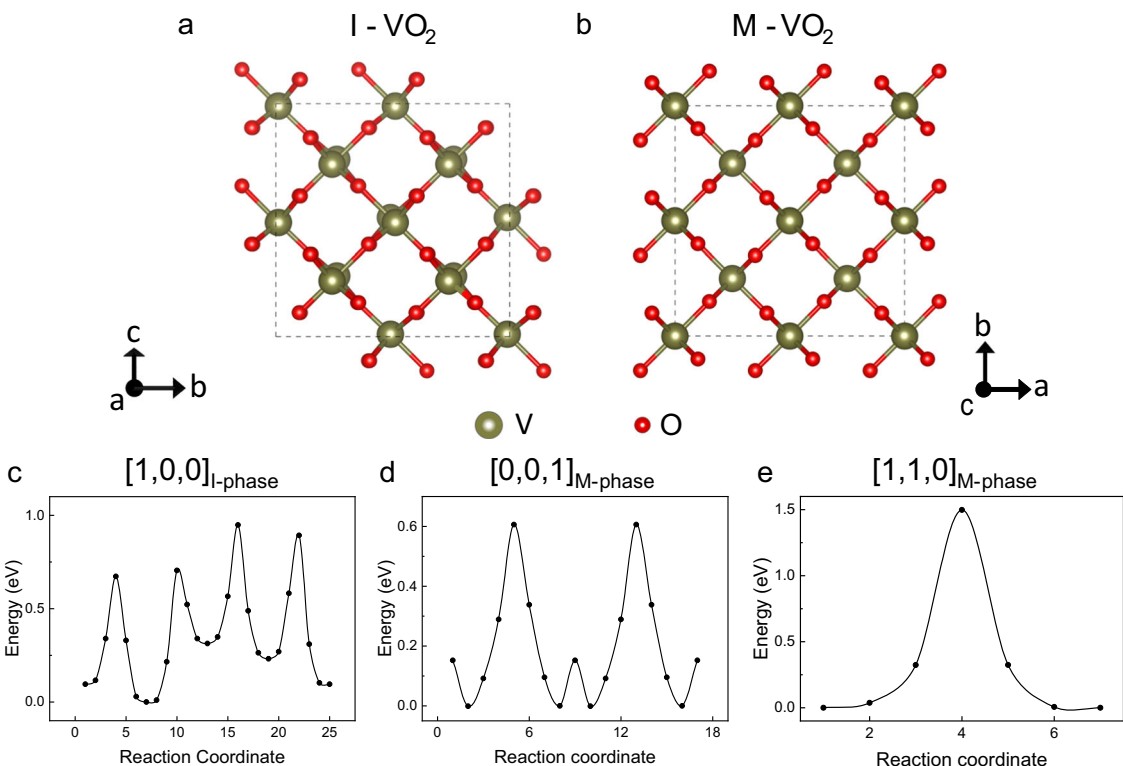

**Fig. 3 | Theoretical calculations for hydrogen diffusion. a, b** Crystal structures of monoclinic (I-phase) and rutile (M-phase) $VO_2$ viewed along the microbeam axis direction: [1,0,0] direction of monoclinic, and [0,0,1] direction of rutile $VO_2$, respectively. **c–e** Calculated lowest-energy path and energy barriers for hydrogen diffusion along the $[1,0,0]_{\text{I-phase}}$, $[0,0,1]_{\text{M-phase}}$, and $[1,1,0]_{\text{M-phase}}$ directions, respectively.

The latter is comparable, within an error range of 0.2 eV, to the activation barrier for hydrogen diffusion along the same direction of rutile $TiO_2$ and $VO_2$ from both theoretical calculations[31–33] and experimental measurements[6,34] including this work. The calculated $E_I^{\text{DFT}}$ in this study which is also consistent with previous calculations[9,35] is in stark contradiction with the observed sharp drop in diffusivity in the low-temperature regime in Fig. 1c that can only be explained by an energy barrier of diffusion on the order of 2 eV.

First-principles calculations have been used to predict the enthalpy ($H$) of $H_xVO_2$ as a function of $x$, as well as its differentiation with respect to $x$ which is the formation energy ($\gamma$) of adding one hydrogen ion into the system[19]. It has been found that the calculated $\gamma_M$ is lower than $\gamma_I$ by $-\Delta\gamma = \gamma_I - \gamma_M$. The value of $-\Delta\gamma$ is found to be ~1 eV when the hydrogen concentration is in the range that we are interested in (comparable to $x_{\text{MIT}}$) (Supplementary Note 4 and Supplementary Table 4). This discontinuity in $\gamma$ across the MIT with respect to hydrogen doping, characterized by a divergence in chemical susceptibility $\alpha = \left(\frac{\partial\gamma}{\partial x}\right)_{T,\sigma}$, leads to a chemical energy penalty of $|\Delta\gamma|$ for a hydrogen ion to diffuse from a M domain into an I domain. This effect is very similar to the mechanical energy penalty of $|\sigma_{\text{MIT}}\Delta\varepsilon|$ that a $VO_2$ microbeam at fixed stress (the so-called "isobaric" condition) needs to pay to transition from the I phase to M phase, also characterized by a divergence in its compressibility $\beta = \left(\frac{\partial\varepsilon}{\partial\sigma}\right)_{T,x}$. Therefore, in the temperature regime of $T_{\text{anneal}} < T_{\text{MIT}}^0$, hydrogen ions first diffuse into I-phase $VO_2$ until its atomic fraction reaches the value of $x_{\text{MIT}}$ at $T_{\text{anneal}}$, so that the surface layer turns into M phase and a M-I domain wall forms. From that point on, subsequent hydrogen ions need to cross the M-I domain wall to further diffuse along the microbeam. In this process, each ion would need to overcome $|\Delta\gamma| \approx 1$ eV on top of the original, pure-phase diffusion energy barrier of $E_I$, resulting in an effective energy barrier of $E = E_I + |\Delta\gamma|$. Taking $E_I \approx E_I^{\text{DFT}} = 0.9$ eV, we

find $E \approx 1.9$ eV, a value consistent with the steeper slope of diffusivity seen in the low-temperature regime in Fig. 1c. We fit to the experimental diffusivity in Fig. 1c with a model that uses an adjustable energy barrier of $E = E_M^{\text{exp}}$ at $T_{\text{anneal}} > T_{\text{MIT}}^0$. When $T_{\text{anneal}} < T_{\text{MIT}}^0$, the model uses the diffusion energy barrier of $E = E_M^{\text{exp}}$ for the M domain and another barrier of $E_I^{\text{exp}}$ for the I domain, respectively. The coexistence of M and I phases results in a graduate slope change from $E_M^{\text{exp}}$ to $E_I^{\text{exp}}$ as temperature decreases (Supplementary Fig. 11). The model yields best-fit values of $E_M^{\text{exp}} \approx 0.8$ eV and $E_I^{\text{exp}} \approx 2.2$ eV, as shown with the red curve in Fig. 4a. The former is close to $E_M^{\text{DFT}}$ and the latter is close to the value of $E_I^{\text{DFT}} + |\Delta\gamma|$, both within an error much smaller than $|\Delta\gamma|$.

To further test the chemically retarded ion diffusion across the M-I domain wall, we compare Pt-catalyzed hydrogen diffusion at the same temperature (37 °C) of two polycrystal thin film samples with similar thickness and crystallinity: an undoped $VO_2$ and a 1.5 at% tungsten-doped $VO_2$ ($W_{0.015}V_{0.985}O_2$). In this way, hydrogen ions diffusing from the surface into the film would have to overcome a M-I domain wall in the former case akin to Fig. 2a, but not in the latter case because $W_{0.015}V_{0.985}O_2$ is already in the M phase at 37 °C. To rule out the influence of tungsten, we also compare hydrogen diffusion in the two thin films above $T_{\text{MIT}}^0$ so that both are in the M phase (Supplementary Fig. 12a). By performing XPS at different stages of the hydrogenation, the hydrogen diffusion speed can be estimated from the growth of the O-H peak at ~532.0 eV in the O-1s core level spectra, as shown in Fig. 4b, c. The initial intensity of this peak, common in $VO_2$ films, may come from adsorbed water molecules and the bottom oxide substrate[28,36]. The peak intensity associated with the O-H bond stays nearly constant for the I-phase $VO_2$, while it rapidly grows for the M-phase $VO_2$ under the same hydrogenation treatment (Supplementary Fig. 12b), implying a much faster incorporation of hydrogen ions in its surface layer. We monitor the conductance of the films ($C$) to

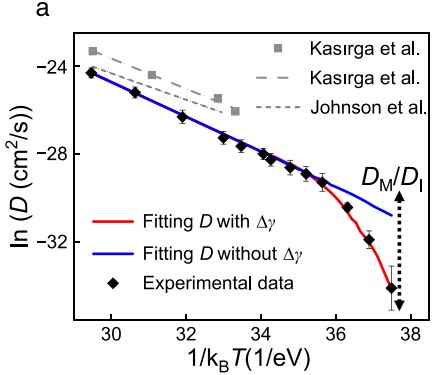

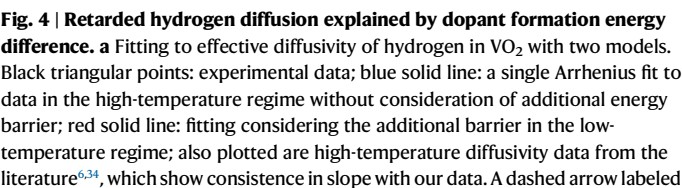

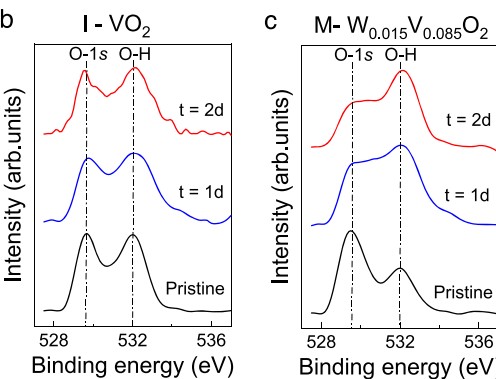

**Fig. 4 | Retarded hydrogen diffusion explained by dopant formation energy difference. a** Fitting to effective diffusivity of hydrogen in $VO_2$ with two models. Black triangular points: experimental data; blue solid line: a single Arrhenius fit to data in the high-temperature regime without consideration of additional energy barrier; red solid line: fitting considering the additional barrier in the low-temperature regime; also plotted are high-temperature diffusivity data from the literature[6,34], which show consistence in slope with our data. A dashed arrow labeled with "$D_M/D_I$" is added, indicating the ratio of hydrogen diffusion in I-phase $VO_2$ retarded from that in M-phase $W_{0.015}V_{0.985}O_2$ film at the same annealing temperature. Error bar: uncertainty of $\ln D$ (see Supplementary Note 2). **b**, **c** XPS O-1$s$ core level of $VO_2$ and $W_{0.015}V_{0.985}O_2$ evolving as a function of hydrogenation days at 37 °C ($T_{anneal} = 37$ °C), showing a much weak hydrogen diffusion in the I phase because it requires to cross M-I domain walls.

estimate the hydrogen diffusion depth, calibrated from the conductivity of pristine ($C_0$) and fully hydrogenated ($C_H$) samples (Supplementary Fig. 13a). Treating the partially hydrogenated films as a pristine film in parallel with a fully hydrogenated layer, the thickness ratio of the M-phase is approximately $(C - C_0)/(C_H - G_0)$ (Supplementary Note 5 and Supplementary Table 5). The diffusion depth in the I-phase $VO_2$ is found to be as low as ~0.5 nm, consistent with the XPS behavior shown in Fig. 4b. In contrast, the diffusion depth of hydrogen in the M-phase $W_{0.015}V_{0.985}O_2$ film is estimated to be about 2 orders of magnitude greater. This large ratio is illustrated in Fig. 4a as a vertical arrow labeled with "$D_M/D_I$", showing consistency with the large retardation observed in the I-phase $VO_2$ microbeams. We also note that the M-I domain wall may pin some of the diffusing hydrogen ions due to lattice discontinuity. However, our Monte Carlo simulations (Supplementary Fig. 14) show that even in the presence of hydrogen ion pile-up at the domain wall, the overall kinetics of hydrogen diffusion along the nanobeam will not be significantly affected, owing to the small thickness and pinning capacity of the domain wall.

## Discussion

As the mechanism does not rely on materials specifics, the chemically retarded ion diffusion in phase-coexisting systems would occur universally in materials where phase transformation is coupled to atomic composition, such as diffusion of dopants and vacancies in $VO_2$[10,37], rare-earth nickelates (e.g. $SmNiO_3$[38–40]) and other complex oxides (e.g. $BaTiO_3$[41], $LaMnO_3$[42]). When diffusing across a domain wall from the high-enthalpy phase to the low-enthalpy phase, each ion needs to overcome an additional energy barrier equal to the difference in formation energy ($\Delta\gamma$) in these two phases, $-\Delta\gamma = -\Delta S \cdot dT_C/dx$, where $dT_C/dx$ is the rate of phase transformation temperature reduced by the doping. $|\Delta\gamma|$ can be estimated by using parameters found in previous reports, and one expects additional energy barrier on the order of $|\Delta\gamma|$ for dopant diffusion in co-existing phases. For example, we predict $|\Delta\gamma|$ to be substantial (~0.4 eV) for oxygen vacancy diffusion that drives the orthorhombic (O) to rhombohedron (R) transition in $LaMnO_3$[43,44]. In metal alloys such as carbon-doped iron (C-Fe)[45] and aluminum-doped titanium (Al-Ti)[46], incorporation of the minority components is known to reduce the bcc-fcc (or hcp) transformation temperature. However, the additional barrier is estimated to be low, on the order of 40 meV, due to the low $\Delta S$ invoked across the phase transformations in these alloy systems (Supplementary Table 6). More broadly speaking, an additional energy barrier will occur when the atom needs to break a

local order parameter to diffuse through the lattice. For example, iron atoms self-diffusing in the ferromagnetic phase of iron may face an additional energy barrier on the order of the magnetic exchange interaction. This insight agrees with earlier experimental observation that the self-diffusion energy of iron in the ferromagnetic state is ~0.4 eV higher than in its paramagnetic state[47].

In summary, hydrogen diffusion in $VO_2$ faces an anomalously high energy barrier when crossing boundaries between two phases. The additional energy barrier is attributed to the volumetric energy penalty that the diffusing ion needs to pay for the imbalance in formation energy in the two phases. The effect is a chemical analogy to superelasticity and diverging compressibility when the phase transition is mechanically driven. The work discovers a new dimension of materials behavior arising from interplay between ion diffusion and phase transformation, two fundamental phenomena central to materials science. It also sheds light on materials design for applications relying on ion diffusion such as hydrogen storage, memristor, and rechargeable ion battery.

## Methods
### Materials preparation
The $VO_2$ single crystals were synthesized using a vapor transport scheme modified from a previously reported method[26]. $V_2O_5$ powder was placed in a quartz boat in the center of a horizontal tube furnace, and evaporated at 900 °C for 2 h, with 6.8 sccm Ar as the carrier gas. The reaction product was collected at 700 °C on an unpolished quartz substrate from downstream of the evaporated sources. The polycrystalline $VO_2$ and W-doped $VO_2$ films were deposited using pulsed laser deposition (PLD). The PLD targets were prepared by mixing $WO_3$ and $V_2O_5$ powders with W: V atomic ratio of 0 or 1.5%. Both films (~100-nm thick) were deposited with the substrate temperature of 560 °C, $O_2$ pressure of 5mTorr, and the PLD laser energy of 330 mJ with pulse frequency of 5 Hz.

### Device fabrication and hydrogen treatment
The $VO_2$ microbeams (200–500-nm thick) were transferred to a new substrate, followed by photolithography and electron beam evaporation of 0.3 nm Platinum on their tip areas as the catalyst. Electron beam evaporation was also utilized to deposit 0.3 nm thick Pt catalyst layer on $VO_2$ and $W_{0.015}V_{0.985}O_2$ thin films. The $VO_2$ microbeams or films deposited with catalyst were placed in a tube furnace with a base pressure of $10^{-3}$ Torr for hydrogenation. The overall pressure was kept at 20 Torr for all hydrogenation experiments.

## Characterization

Raman spectra were measured by a Raman spectrometer (Renishaw Inc.) with a 488 nm laser as the excitation source and a laser power of 0.01 mW. SEM-EBSD (FEI Quanta 3D FEG) was utilized to determine the single crystallinity and crystal orientation of $VO_2$ microbeams. XPS (Physical Instruments 5600/5800; Al Ka, 1486.7 eV) measurements were performed to examine the chemical states of the $VO_2$ microbeams and films. For TEM characterization, single-crystalline $VO_2$ beams were dry transferred from the growth substrate onto TEM grids using a probe station. High-resolution transmission electron microscopy, as well as the corresponding energy dispersive X-ray spectroscopy (EDS) were carried out on a ThemIS microscope at accelerating voltage of 300 kV in National Center for Electron Microscopy (NCEM). The reference diffraction patterns were simulated by Single Crystal 4.

## First-principles calculations

The calculations based on density functional theory (DFT) were carried out to explore the dynamic properties of H atoms in the vanadium dioxide, using the Vienna ab initio Simulation Package (VASP) code[48]. The Core electrons were described by pseudopotentials generated from the projector augmented wave method and the exchange and correlation terms were described using general gradient approximation (GGA) in the scheme of Perdew-Burke-Ernzerhof (PBE)[49,50]. The cut-off energy for the plane-wave basis was 450 eV. The DFT + U method was employed to optimize the structure, where $U$ and $J$ were chosen to be 4.0 eV and 0.68 eV, respectively. The $V_4O_8$ unit cell with one hydrogen atom doped was used to study the diffusion barrier of hydrogen in $VO_2$. $6 \times 8 \times 8$ k-point meshes were used to sample the Brillouin zone. Lattice constants and internal coordinates were fully optimized until all forces on the free ions were converged to 0.01 eV/Å. Climbing Image Nudged Elastic Band (CI-NEB) method[51] was used to find the minimum energy paths and the transition states.

## Data availability

The data generated in this paper have been deposited in Zenodo under accession code (https://zenodo.org/records/12242975). All other data are available from the corresponding author upon request.

## Code availability

The codes used in this paper have been deposited in Zenodo under accession code (https://zenodo.org/records/12242975). All other codes are available from the corresponding author upon request.

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

## Acknowledgements

The work was supported by the U.S. Department of Energy, Office of Science, Office of Basic Energy Sciences, Materials Sciences and Engineering Division under contract no. DE-AC02-05CH11231 (Electronic Materials program). Z.W. acknowledges the support by Innovation Program for Quantum Science and Technology (2021ZD0303303), the CAS Project for Young Scientists in Basic Research (YSBR-005), the National Natural Science Foundation of China (22025304, 22033007). J. Wan and H.Z. acknowledge the support of the U.S. Department of Energy, Office of Science, Office of Basic Energy Sciences (BES), Materials Sciences and Engineering Division under Contract No. DE-AC02-05-CH11231 within the in-situ TEM program (KC22ZH). We are grateful to Dr. Kaichen Dong, Dr. Naoki Higashitarumizu, and Yinhe Wang for experimental and calculation assistance.

## Author contributions

Y.C. and J. Wu conceived this project. Z.W. and J.J. performed DFT calculations and analysis. Y.C. synthesized the single-crystalline VO₂ microbeams, deposited VO₂ thin films and fabricated the devices under the support of A.J. and J.A. Y.C. completed measurement, with the assistance of R.G. and J.L. J. Wan and H. Z. contributed to the TEM imaging. Y.C. and J. Wu analyzed the results. All authors discussed and contributed to the preparation of the manuscript.

## Competing interests

The authors declare no competing interests.
