## [Peer Review File · Nature Communications]

Ion Diffusion Retarded by Diverging Chemical SusceptibilityREVIEWER COMMENTS

Reviewer #1 (Remarks to the Author):

The present manuscript reports anomalously high energy barrier for dopant diffusion in co-existing phases across the metal to insulator transition point of VO₂, and attributed this behavior to the volumetric energy penalty that the diffusers need to pay for the reduction of latent heat. In general, the manuscript is well written and the conclusion is rather interesting. Nevertheless, I have the following suggestions for further improvements.

1) Figure 1c, how the error bar was determined when calculating D?

2) Also for Figure 1c, another thing is that it seems that D is smaller at the insulating phase region approaching to the critical temperature but the D-T tendency seems smooth in the region associated with the metallic phase approaching to the critical temperature. I was wondering how the critical temperature (TMIT) should be determined, since it is hard to tell whether the metallic and insulating phase 'should' co-exist in across TMIT or just at the temperature region slightly below TMIT? This needs to be further clarified.

3) The experiments as demonstrated in Figure 2 was well designed, and in the present case the hydrogenation seems to reduce the resistivity of VO₂ that stabilizes the metallic phase. Nevertheless, the hydrogenation of VO₂ can also trigger an insulating state under mild hydrogenation temperatures and high hydrogen concentration (e.g., Nat. Mater. 2016, 15, 1113, or J. Phys. Chem. Lett. 2022, 13, 8078), in which situation higher amount of hydrogen will be incorporated within the material. I was wondering whether the mixing phase adjacent to the metal to insulator transition temperature will affect D in this case.

4) It seems that the less effective hydrogen diffusion within the mixing phase region was caused by the additional energy as caused by the metallicity of VO₂ via hydrogenation. Nevertheless, I doubt whether such energy should be considered to be analogous to the latent heat as released when varying temperature across TMIT. Of course this may be correct, but I suggest to be more careful in making a universal conclusion from that perspective.

5) Comparable results to Figure 1c should be also demonstrated for W doped VO₂, in which situation the magnitude of TMIT is reduced via substituting V by transition metal with higher valance. Whether this can shift the dash line as shown in Figure 1c leftwards will to some

extent testify the universality of the present conclusion.

6) One confirming experiment worthy to be considered in the future (not necessarily to be done now for this work) is associated with the hydrogen diffusion within rare-earth nickelates. From the one side the latent heat across TMIT is much smaller compared to VO₂, while the hydrogenation triggered reduction in G should be comparable. From the other side, the resistivity of rare-earth nickelates elevates abruptly, the tendency of which is just the opposite compared to the present one. This will be a perfect supplementary experiment to the present work.

Reviewer #2 (Remarks to the Author):

In this work, the authors provided the effect of second derivative of Gibbs free energy on hydrogen diffusion in VO₂. It reported hydrogen faced three times higher energy barrier and over one order of magnitude lower diffusivity when crossing a metal-insulator domain wall in VO₂. Furthermore, they employed the first-principles calculations to reveal the hydrogen diffusion along various axes. However, after a comprehensive evaluation of this work, I think it can't provide profound scientific meaning, also can't attract broad attention.

1) As shown in Formula (1), it seems that the Gibbs free energy is linear with the hydrogen atomic fraction (x). However, the doping of hydrogen in VO₂ could induce a metallic or insulator state depending on the concentration of hydrogen (Nat. Mater. 15, 1113, 2016). Obviously, the provided module can't illustrate the effect of high concentration hydrogen on VO₂.

2) The diffusion energy barriers of hydrogen along various axes of VO₂ were calculated, while the specific diffusion routes should be present in Fig. 3.

Reviewer #3 (Remarks to the Author):

In the present manuscript, one chemical analogy was proposed to display the singularity of the second derivatives of Gibbs free energy across first-order phase transitions, leading to an anomalously high energy barrier for dopant diffusion in co-existing phases within the hydrogenated VO₂ single crystal microbeams and polycrystalline films. This work also applied the chemical susceptibility to analyze other co-existing phase systems. In general

the manuscript is interesting but the following concerns must to be addressed:

- 1) The author attempts to introduce the contribution of hydrogen occupancy in the driving force of phase transition and explains the singularity caused by this term. In the first paragraph of page 9 of the manuscript, the author points out: "...leads to a chemical energy penalty of $|\Delta\gamma|$ for a hydrogen ion to diffuse from an M-domain into an I-domain. This effect is very similar to the mechanical energy penalty of $|\sigma_{MIT}\Delta\epsilon|$...". However, the chemical energy dependency caused by hydrogen might be essentially considered as the stress effect itself. For materials with phase transition, the local chemical stress caused by doping often exhibits the same effect as macroscopic stress, such as biaxial stress caused by the substrate in nickelates (Matter, 2020, 2, 1) and isotopic doping effects in conductive polymers (J. Chem. Soc., Perkin Transaction 2, 1995). In Table.S3, it can be seen that the lattice expansion of the hydrogenated rutile phase and intrinsic rutile phase reaches 4%. Therefore, it is suggested that the author increase the discussion on local chemical pressure to exclude the influence of other factors on the analogy of chemical magnetization.
- 2) Fig.S7 is not mentioned in the main text. The accompanying issue is that this article lacks a sufficient explanation of the structural information, morphology, and orientation of synthesized microbeams in the main manuscript.
- 3) Fig.2(b) lacks of detailed legend to distinguish between hollow triangles and solid circles.
- 4) The ex-situ observation of Raman spectra and fitting of diffusion coefficients for hydrogen-induced MIT may overlook some potential factors, such as the chemical gradient introduced by the non-uniform distribution of hydrogen atoms along the microbeams. The influence of this part on the hydrogen diffusion coefficient is usually not significant. But when more hydrogen atoms are pinned onto the domain wall (due to the diffusion barriers, in the author's context), the blocking effect of this part of the chemical gradient is not considered. I suggest the author to further discuss to clarify the effects of enthalpy and formation energy.
- 5) In XPS analysis, the integrated area of the peak may be more convincing than the peak intensity. This can be achieved through more detailed peak fitting.
- 6) Lack of discussion on the difference in hydrogen diffusion rates between pristine tungsten-doped samples and pristine VO₂ samples. The effect of tungsten dopant on hydrogen diffusion may affect the independence of domain wall analysis.

7) The analysis of Fig.1(d) appears in multiple places in the text, which is relatively scattered and not conducive to the coherence of the discourse. The relevant paragraphs on page 7 can be integrated with those on page 9.

8) From a microscopic perspective, will hydrogen atoms continuously cross the M-I domain wall barrier during diffusion? This may be related to whether hydrogen is the sole driving force behind the M-I transition when $T < T_{MIT}$. The lattice expansion caused by electron doping may drive the local transformation of other I-phases. Therefore, hydrogen may be discontinuous across domain walls. The exploration of this mechanism may require deeper calculations and more powerful characterization techniques in the future.

9) I am curious about the result of simultaneous hydrogenation on both sides of a finite beam. For a sufficiently uniform microbeam, hydrogen is expected to symmetrically promote the formation of the M-phase under the same conditions. When the domain walls on both sides are very close, the phase transition behavior of the central I phase may become anomalous. When any hydrogen atom overcomes a chemical barrier, it will cause a sudden collapse of the domain wall on one side of the phase. This may trigger novel physical phenomena. In future work, the author can continue to explore in depth.

10) Concerning the hydrogen interaction with correlated or phase transition materials, other systems than VO₂ should be also mentioned, and whether analogous effects are expected should be also discussed (e.g., Nat commun 2019, 10, 694; Adv Mater 2020, 1905060)

Reviewer #4 (Remarks to the Author):

The authors introduced chemical susceptibility as a thermodynamic parameter to show that it diverges near a first-order phase transition. They used hydrogen-doped vanadium dioxide (VO₂) as a model system and based their findings on their experimental work on VO₂ nano-beams hydrogenated via spill-over. The manuscript brings a new perspective to hydrogen doping and its effects on VO₂. Moreover, the findings reported in the manuscript have the potential to impact related research fields. However, I think further clarification on various aspects of the study is required before I can recommend the publication of the manuscript.

Please find my comments below:

1. Although historically 67 C is a widely accepted value for T_c of MIT in VO₂, Park et al.

Nature (2013) paper sets this value to be 65 C. I think this value should be adopted, especially in the context of this paper, as different insulating phases are disregarded. Could the authors either adopt 65 C as T_c or provide reasoning regarding why they use 67 C, which is a relic from old studies on bulk or epitaxial films?

2. Could the authors comment on the presence of M1 and M2 phases in the insulating state of VO₂ and how this might affect their conclusions? As they relieved the nonuniform strain due to the substrate adhesion, their crystals should be in either M1 or M2 phase.

3. The authors assumed that hydrogen fraction in VO₂ (x) causes a linear change in the T_{mit} . This is backed up by the ab initio calculations in Ref.19 (in the manuscript) as well as from the gradual shift of the M-I domain wall gradually shift to the other tip when heated towards T_{mit}^0 . However, although samples doped above 100 C for a short duration exhibit such behavior, samples doped at lower temperatures for longer durations show less sharp M-I boundaries, as in Ref. 6. This is also somewhat evident in Figure S6b, as from 2 to 8 to 16 hours the resistance change doesn't seem to follow a sharp boundary. Could the authors explain how this faint M-I boundary fits into the thermodynamic picture they propose?

4. Although W-doped measurements somewhat serve that purpose, could the authors elaborate more on the spillover rate of the Pd catalyst at low temperatures, i.e. below T_c ? One peculiar aspect related to the spillover rate is much long durations required for hydrogen doping of W-doped VO₂ as compared to the undoped case above T_c . This brings up the issue that if the Pd spillover is slower at lower temperatures along with the lower diffusion coefficient of the insulating phases, there should be an additional correction to the calculated γ factor.

5. Relevant to point 4, in Note S4, the authors mention how electrical measurements on tungsten-doped polycrystalline samples are used to measure the diffusion depth. However, no conductivity data is presented. It would be helpful to include the conductivity results in the supporting information.

6. A scale bar to the OM inset of Fig2b would be helpful.

Responses to referees:

Reviewer #1:

The present manuscript reports anomalously high energy barrier for dopant diffusion in co-existing phases across the metal to insulator transition point of VO₂, and attributed this behavior to the volumetric energy penalty that the diffusers need to pay for the reduction of latent heat. In general, the manuscript is well written and the conclusion is rather interesting. Nevertheless, I have the following suggestions for further improvements.

Thank you for your encouraging and constructive comments!

Comment #1: Figure 1c, how the error bar was determined when calculating D ?

Response #1: In Fig. 1(c), following $\ln D = \ln(L_M^2/t)$, the error bar is determined mathematically: $\Delta(\ln D) = \Delta D/D = 2L_M\Delta L_M/Dt$, and ΔL_M comes from the measurement of L_M . Details of determining ΔL_M and $\Delta(\ln D)$ have been added as Note S2 in the supplementary information.

Comment #2: Also for Figure 1c, another thing is that it seems that D is smaller at the insulating phase region approaching to the critical temperature but the D - T tendency seems smooth in the region associated with the metallic phase approaching to the critical temperature. I was wondering how the critical temperature (T_{MIT}) should be determined, since it is hard to tell whether the metallic and insulating phase ‘should’ co-exist in across T_{MIT} or just at the temperature region slightly below T_{MIT} ? This needs to be further clarified.

Response #2: This is an insightful comment and points out the uniqueness and complexity of this system:

- (1) T_{MIT}^0 of the samples is determined to be sharp at 67°C by optical inspection before any hydrogen treatment. It’s mentioned on page 5 of the main text and shown in Fig. S3(b). The dashed line in Fig. 1(c) corresponds to $T = T_{MIT}^0$.
- (2) The M and I phases coexist as long as $T_{anneal} < T_{MIT}^0$, not only in the region slightly below T_{MIT}^0 , which has been proved in Fig. S4 ($T_{anneal} \ll T_{MIT}^0$). The offset of T_{MIT}^0 and the kink of the D - T curve is explained in the Supplementary information and summarized here. As shown in Fig.1(b) and also below, when T_{anneal} is below but close to T_{MIT}^0 , the threshold hydrogen fraction for making the MIT is $x_{MIT} \sim 0$, hence nearly the entire beam will be in M phase, so the measured D will follow the metallic D - T curve. On the contrary, if T_{anneal} is much lower than T_{MIT}^0 , x_{MIT} is much higher (approaching x_{MIT}^0), the hydrogen diffusion will slow down at the M-I phase boundary, the overall D then follows the insulating D - T curve.

This conclusion has been mentioned on page 9 of main text and proved by the simulations in Fig. S11(b). The captions of Fig. S11(b) have been modified to clarify.

Comment #3: The experiments as demonstrated in Figure 2 was well designed, and in the present case the hydrogenation seems to reduce the resistivity of VO₂ that stabilizes the metallic phase. Nevertheless, the hydrogenation of VO₂ can also trigger an insulating state under mild hydrogenation temperatures and high hydrogen concentration (e.g., Nat. Mater.2016, 15, 1113, or J. Phys. Chem. Lett. 2022,13,8078), in which situation higher amount of hydrogen will be incorporated within the material. I was wondering whether the mixing phase adjacent to the metal to insulator transition temperature will affect D in this case.

Response #3: In our case of hydrogen diffusion in single-crystalline VO₂ microbeams, high concentration (high-H) insulating state (HVO₂) has been ruled out by a few experimental evidence:

- (1) If there was a high-H insulating phase, it should appear at the tip area of microbeams (where H fraction is the highest), and easily differentiated with the metallic phase under optical microscope (Sci. Adv. 2019;5: eaav6815). This should result in an I-M-I domain distribution. However, it has never been observed in our experiment, from 37°C to 120°C.
- (2) The lattice expansion of our hydrogen doped VO₂ is ~4% (by TEM), much smaller than the reported values for the high-H insulating phase (~10% in Nat. Mater.2016, 15, 1113, or J. Phys. Chem. Lett. 2022,13,8078). This confirms our lower H doping concentrations in the case of VO₂ microbeams.
- (3) The high-H insulating phase was found in VO₂ thin films where the thickness (diffusion path) is on the nanometer scale. For VO₂ beams in our case, the diffusion is along the beam direction where the length of diffusion path is over micrometers or longer scale. It will take much longer time (or higher temperature) for hydrogen to reach the end of the diffusion channel ($L \sim \sqrt{Dt}$) to get saturated – only after that, diffusing hydrogen ions will begin to build up toward the ~ 10% concentration level. Therefore, we do not enter that high-H scenario.

Comment #4: It seems that the less effective hydrogen diffusion within the mixing phase region was caused by the additional energy as caused by the metallicity of VO₂ via hydrogenation. Nevertheless, I doubt whether such energy should be considered to be

analogous to the latent heat as released when varying temperature across T_{MIT} . Of course this may be correct, but I suggest to be more careful in making a universal conclusion from that perspective.

Response #4: We use the analogy to make the effect more easily understandable to readers, as when the phase transition is triggered by increasing generalized “force”: temperature, force, or chemical doping, the corresponding generalized “work” is latent heat, work, or chemical energy. We understand the reviewers’ concern due to the abstractness of chemical energy, but from basic thermodynamics we believe it is a good analogy considering the consistency between the analogy and DFT calculations as well as experiments. Nevertheless, we treated the reviewer’s comment seriously and modified our wording on page 4 in the main text.

Comment #5: Comparable results to Figure 1c should be also demonstrated for W doped VO₂, in which situation the magnitude of T_{MIT} is reduced via substituting V by transition metal with higher valance. Whether this can shift the dash line as shown in Figure 1c leftwards will to some extent testify the universality of the present conclusion.

Response #5: As the reviewer suggests, results on hydrogen diffusion in W-doped VO₂ may exclude other factors and make the conclusion universal if the dashed line and the data points would move rightwards (toward lower T). That was exactly why we have performed experiments to compare hydrogen diffusivity in VO₂ and W-doped VO₂ (Fig. 4(a) and Fig. S13(a)).

To test this and rule out unwanted influence of tungsten in H diffusion (other than simply shifting T_{MIT}) in analysis in Fig.4(a), we have added an experiment (Fig.S12(a)) to qualitatively compare the hydrogen diffusivity in VO₂ and W-doped VO₂ at 90°C, a temperature when both are metallic. As shown below in (a), the diffusion rate of hydrogen in M-VO₂ and M-W_{0.015}V_{0.985}O₂ at the same, high temperature is similar to each other. However, in (b), the diffusion is retarded (orders of magnitude) in I-VO₂ compared to M-W_{0.015}V_{0.985}O₂ at the same, low temperature (37°C). Therefore, it is the phase being M or I that causes the diffusion retardation. The retarded diffusivity in VO₂ compared with W_{0.015}V_{0.985}O₂ indicates that the curve in Fig. 1(c) indeed would shift rightwards for W_{0.015}V_{0.985}O₂ compared to VO₂. A related statement has been added to the main text (in front of ‘By performing XPS’ on page 10 of the main text).

Comment #6: One confirming experiment worthy to be consider in the future (not necessarily to be done now for this work) is associated with the hydrogen diffusion within rare-earth nickelates. From the one side the latent heat across T_{MIT} is much smaller compared to VO₂, while the hydrogenation triggered reduction in G should be comparable. From the other side, the resistivity of rare-earth nickelates elevate abruptly, the tendency of which is just the opposite compared to the present one. This will be a perfect supplementary experiment to the present work.

Response #6: Thank you for the suggestion: this comment addresses the universality of this effect in the field of material science, aligning well with what we have proposed in the discussion session. Considering the MIT of rare-earth nickelates as well as the hydrogen induced phase transition induced by d-band electron correlation, a similar effect may exist when hydrogen diffuses within rare-earth nickelates.

However, as the reviewer suggests, rare-earth nickelates turn into a more insulating state (e.g. SmNiO₃, Nature Communications 5, 4860 (2014)) with hydrogen doping, opposite to the VO₂ system. A short discussion and a few references related to this topic has been added into the main text (Discussion Section), and we'll consider doing the experiment in the future.

Reviewer #2:

In this work, the authors provided the effect of second derivative of Gibbs free energy on hydrogen diffusion in VO₂. It reported hydrogen faced three times higher energy barrier and over one order of magnitude lower diffusivity when crossing a metal-insulator domain wall in VO₂. Furthermore, they employed the first-principles calculations to reveal the hydrogen diffusion along various axes. However, after a comprehensive evaluation of this work, I think it can't provide profound scientific meaning, also can't attract broad attention.

Thank you for your comments! We believe we fully addressed your two comments below.

Comment #1: As shown in Formula (1), it seems that the Gibbs free energy is linear with the hydrogen atomic fraction (x). However, the doping of hydrogen in VO₂ could induce a metallic or insulator state depending on the concentration of hydrogen (Nat. Mater.15, 1113, 2016). Obviously, the provided module can't illustrate the effect of high concentration hydrogen on VO₂.

Response #1: This comment is similar to the Comment #3 from Reviewer #1. Unlike in VO₂ thin films, in hydrogen diffusion along the axial direction of our single-crystalline VO₂ microbeams, the high concentration (high-H) insulating state of VO₂ (HVO₂) does not occur:

- (4) If there was a high-H insulating phase, it should appear at the tip area of microbeams (where H fraction is the highest), and easily differentiated with the metallic phase under optical microscope (Sci. Adv. 2019;5: eaav6815). This should result in an I-M-I domain distribution. However, it has never been observed in our experiment, from 37°C to 120°C.
- (5) The lattice expansion of our hydrogen doped VO₂ is ~4% (by TEM), much smaller than the reported values for the high-H insulating phase (~10% in Nat. Mater.2016, 15, 1113, or J. Phys. Chem. Lett. 2022,13,8078). This confirms our lower H doping concentrations in the case of VO₂ microbeams.
- (6) The high-H insulating phase was found in VO₂ thin films where the thickness (diffusion path) is on the nanometers or sub-micron scale. For VO₂ beams in our case, the diffusion is along the beam direction where the length of diffusion path is over many micrometers or longer scale. It will take much longer time (or much higher temperature) for hydrogen to reach the end of the diffusion channel ($L \sim \sqrt{Dt}$) to get saturated – only after that, diffusing hydrogen ions will begin to build up toward the ~ 10% concentration level. Therefore, we do not enter that high-H scenario. This is simply akin to the fact that rain can easily flood a pond but cannot easily flood a lake.

Therefore, it is safe to consider only insulating VO₂ and metallic H_xVO₂, as we are limited to the small H fraction scenario.

Comment #2: The diffusion energy barriers of hydrogen along various axes of VO₂ were calculated, while the specific diffusion routes should be present in Fig. 3.

Response #2: Thanks for the suggestion to improve the quality. The specific diffusion routes have been added to Fig. S10. Below is an example (along [100]_{I-phase, monoclinic} channel).

Reviewer #3 (Remarks to the Author):

In the present manuscript, one chemical analogy was proposed to display the singularity of the second derivatives of Gibbs free energy across first-order phase transitions, leading to an anomalously high energy barrier for dopant diffusion in co-existing phases within the hydrogenated VO₂ single crystal microbeams and polycrystalline films. This work also applied the chemical susceptibility to analyze other co-existing phase systems. In general the manuscript is interesting but the following concerns must to be addressed:

Thank you for your comments! We address your comments one by one below.

Comment #1: The author attempts to introduce the contribution of hydrogen occupancy in the driving force of phase transition and explains the singularity caused by this term. In the first paragraph of page 9 of the manuscript, the author points out: "...leads to a chemical energy penalty of $|\Delta\gamma|$ for a hydrogen ion to diffuse from an M-domain into an I-domain. This effect is very similar to the mechanical energy penalty of $|\sigma_{MIT}\Delta\varepsilon|$ ". However, the chemical energy dependency caused by hydrogen might be essentially considered as the stress effect itself. For materials with phase transition, the local chemical stress caused by doping often exhibits the same effect as macroscopic stress, such as biaxial stress caused by the substrate in nickelates (Matter, 2020, 2, 1) and isotopic doping effects in conductive polymers (J. Chem. Soc., Perkin Transaction 2, 1995). In Table.S3, it can be seen that the lattice expansion of the hydrogenated rutile phase and intrinsic rutile phase reaches 4%. Therefore, it is suggested that the author increase the discussion on local chemical pressure to exclude the influence of other factors on the analogy of chemical magnetization.

Response #1: The comment states that the effect of local chemical force might be considered as a stress effect itself. The statement itself is correct. In the VO₂ system, external stress (such as substrate-imposed strain) and hydrogen doping (which includes a chemical stress effect) both trigger the phase transition.

- (1) External stress is absent for our VO₂ microbeams which were dry transferred from the growth substrate to and loosely sitting on a new substrate, as confirmed by the sharp phase transition shown in Fig.S3(b). The absence of external stress in this type of unbent, transferred VO₂ beams has been extensively discussed in, for example, Nature Nanotechnology, 4, 732(2009).
- (2) When the formation energy is calculated, the resultant H_xVO₂ lattice is found to have expanded. The internal chemical stress is, therefore, already included in the total energy

calculation and analysis of H diffusion in VO₂ (Phys. Chem. Chem. Phys., 2015, 17, 20998). A few sentences have been added to Note S4 to acknowledge and illustrate this point.

Comment #2: Fig.S7 is not mentioned in the main text. The accompanying issue is that this article lacks a sufficient explanation of the structural information, morphology, and orientation of synthesized microbeams in the main manuscript.

Response #2: We apologize for missing this information. Figure S7 (SEM and EBSD) includes the structural info and morphology of the microbeams. The reason why it was not mentioned in the previous version of main manuscript is that the structural info is not the focus of this work; it is included in Fig.S7 for completion purpose. The fact that VO₂ prefers to grow along [001]_{Rutile} ([100]_{Monoclinic}) with facets of [110]_{Rutile} ([122]_{Monoclinic}) has been reported in earlier work (e.g., Applied Physics Letters 100, 103111 (2012)). The main text in page 7 is now revised to connect the main text and the supplementary information.

Comment #3: Fig.2(b) lacks of detailed legend to distinguish between hollow triangles and solid circles.

Response #3: Hollow triangles and solid circles are data points from two independent samples with the same treatment to prove the repeatability of our measurements. Detailed information has now been added to the captions.

Comment #4: The ex-situ observation of Raman spectra and fitting of diffusion coefficients for hydrogen-induced MIT may overlook some potential factors, such as the chemical gradient introduced by the non-uniform distribution of hydrogen atoms along the microbeams. The influence of this part on the hydrogen diffusion coefficient is usually not significant. But when more hydrogen atoms are pinned onto the domain wall (due to the diffusion barriers, in the author's context), the blocking effect of this part of the chemical gradient is not considered. I suggest the author to further discuss to clarify the effects of enthalpy and formation energy.

Response #4: We suppose that the reviewer is talking about potential high hydrogen concentration pile-up at the domain wall as shown below. A localized high concentration is associated with a localized dip of potential (formation energy), which might be possible at the domain wall.

We did Monte-Carlo simulation to test whether the “pinning” or pile-up of hydrogen atoms will affect the overall diffusion kinetics. According to the simulation below (vertical direction is hydrogen concentration in a.u., horizontal direction is distance along the diffusion direction), the localized hydrogen concentration pile-up is found to be related to the potential dip as $\chi = \chi_0 \exp(-\Delta E/kT)$, consistent with Boltzmann distribution. However, the existence of the dip and pile-up at the domain wall does not affect the overall diffusion of ions: curves with different ΔE values collapse onto each other in regions outside the dip region.

In conclusion, even if the hydrogen ions pile up at the M-I domain walls, they won't significantly affect the overall diffusion kinetics, hence our conclusion still holds. However, this analysis does not rule out the hydrogen pinning effect itself; indeed, the domain boundary is an interesting entity worth further investigation as a separate project. We appreciate the reviewer's comment on this! We have added a statement in the main text to address this at the end of discussion of Fig.4.

Comment #5: In XPS analysis, the integrated area of the peak may be more convincing than the peak intensity. This can be achieved through more detailed peak fitting.

Response #5: This comment is useful to improve the details. The multi-peak fitting is added to Fig. S12. The area info is included in the captions of Fig. S12.

Comment #6: Lack of discussion on the difference in hydrogen diffusion rates between pristine tungsten-doped samples and pristine VO₂ samples. The effect of tungsten dopant on hydrogen diffusion may affect the independence of domain wall analysis.

Response #6: This is similar to Comment #5 of Reviewer #1. To rule out the influence of tungsten, we have added the experiment (Fig.S12(a)) to qualitatively compare hydrogen diffusivity in VO₂ and W-doped VO₂ at 90°C, a temperature when both are metallic phase. In Fig. S12(a), as the reviewer suggests, the multi-peak fitting shows that the diffusion rate of hydrogen in M-VO₂ and M-W_{0.015}V_{0.985}O₂ at the same high temperature is close to each other. However, the diffusion is much retarded (by orders of magnitude) in I-VO₂ compared to M-

$W_{0.015}V_{0.985}O_2$ at a low temperature (37°C). Therefore, it is the phase being M or I that causes the diffusion retardation, not the tungsten dopants themselves. To clarify this, text has been added to the main manuscript (in front of ‘By performing XPS’ on page 10 of main text). Fig.S12 as shown below is also added into the Supplementary Information.

Comment #7: The analysis of Fig.1(d) appears in multiple places in the text, which is relatively scattered and not conducive to the coherence of the discourse. The relevant paragraphs on page 7 can be integrated with those on page 9.

Response #7: Thanks for the suggestion on the writing; we suppose the reviewer is talking about the analysis of the Arrhenius plot of diffusivity (Fig. 1(c)). We understand that the analysis on these two pages may look duplicated because the numbers are repeated. However, these numbers are obtained under different logic: the paragraph on page 7 talks about the initial analysis, extracting the slope from the plot mathematically (define the problem), while the paragraph in page 9 talks about the values from numerical simulations of diffusion based on the presented theory (solve the problem). Therefore, we decide to keep the plots as they are to ensure a good logic flow for the readers to follow.

Comment #8: From a microscopic perspective, will hydrogen atoms continuously cross the M-I domain wall barrier during diffusion? This may be related to whether hydrogen is the sole driving force behind the M-I transition when $T < T_{MIT}$. The lattice expansion caused by electron doping may drive the local transformation of other I-phases. Therefore, hydrogen may be discontinuous across domain walls. The exploration of this mechanism may require deeper calculations and more powerful characterization techniques in the future.

Response #8: This is a great question! Typically the discontinuity of particles' concentration is associated with a stepped potential (e.g. a barrier) such as 2d electron gas in quantum wells (Phys. Rev. B **54**, 10609).

(1) We did Monte-Carlo simulation to simulate particles' diffusion with a barrier (ΔE). It turns out that if there was a discontinuity ($x(\text{left}) \neq x(\text{right})$) due to the barrier, $x(\text{left})/x(\text{right}) = \exp(-\Delta E/kT)$. When $\Delta E = 0.06\text{eV}$ only, $x(\text{left})/x(\text{right}) \sim 8$, which is

contradictory to the experimental results shown in point (2) below.

- (2) If there was a large discontinuity at the domain boundary at T_{anneal} , while cooling down the microbeams to room temperature, the domain boundary should remain nearly the same location, because the threshold concentration doesn't vary drastically (e.g., $x_{\text{MIT}}(41^\circ\text{C}) \sim 0.5x_{\text{MIT}}(20^\circ\text{C})$). In contrast, when the microbeam annealed at 41°C is cooled down to room temperature (optical images shown below), the domain boundary gradually moves toward the high-concentration, left-hand side, showing gradual variation of hydrogen concentration (x) along the diffusion path; therefore, there is no strong discontinuity in x along the beam direction.

As the reviewer suggests, to resolve possible hydrogen concentration discontinuity at these scales in the future, more power techniques may be used, such as nano-SIMS (Annu. Rev. Anal. Chem. 2020. 13:273–92).

Comment #9: I am curious about the result of simultaneous hydrogenation on both sides of a finite beam. For a sufficiently uniform microbeam, hydrogen is expected to symmetrically promote the formation of the M-phase under the same conditions. When the domain walls on both sides are very close, the phase transition behavior of the central I phase may become anomalous. When any hydrogen atom overcomes a chemical barrier, it will cause a sudden collapse of the domain wall on one side of the phase. This may trigger novel physical phenomena. In future work, the author can continue to explore in depth.

Response #9: This comment provides a future direction for discovering novel phenomena in the present system as a useful material platform. We appreciate the reviewer's suggestion! This further highlights the importance and broad impact of our work for future studies.

Comment #10: Concerning the hydrogen interaction with correlated or phase transition materials, other systems than VO₂ should be also mentioned, and whether analogous effects are expected should be also discussed (e.g., Nat commun 2019, 10, 694; Adv Mater 2020, 1905060)

Response #10: This comment points to possible universality of this effect in the field of d-electron correlated materials. Considering the MIT of rare-earth nickelates as well as the hydrogen induced phase transition, a similar effect may exist when hydrogen diffuses within rare-earth nickelates.

Rare-earth nickelates turn into a more insulating state with hydrogen doping (e.g. SmNiO₃, Nature Communications 5, 4860 (2014)), opposite to the VO₂ system. In the works suggested by the reviewer, more complex behavior has been introduced (e.g. reduction of resistance by a factor of 50 with high hydrogen concentration). However, calculation on formation energy of hydrogen or how hydrogen affects T_{MIT} is absent, therefore, more work is needed to make a quantitative comparison. Qualitatively, if hydrogen enhances the insulating state, the energy barrier for hydrogen to be doped into rare-earth nickelates should be higher in the metallic state. In the future, hydrogen diffusion in SmNiO₃ around T_{MIT} (~400K) will be a good future experiment to test this general effect. A brief discussion and references mentioned above have been added into the main text (Discussion).

Reviewer #4 (Remarks to the Author):

The authors introduced chemical susceptibility as a thermodynamic parameter to show that it diverges near a first-order phase transition. They used hydrogen-doped vanadium dioxide (VO₂) as a model system and based their findings on their experimental work on VO₂ nano-beams hydrogenated via spill-over. The manuscript brings a new perspective to hydrogen doping and its effects on VO₂. Moreover, the findings reported in the manuscript have the potential to impact related research fields. However, I think further clarification on various aspects of the study is required before I can recommend the publication of the manuscript.

Thank you for your positive and encouraging comments!

Please find my comments below:

Comment #1: Although historically 67 C is a widely accepted value for T_c of MIT in VO₂, Park et al. Nature (2013) paper sets this value to be 65 C. I think this value should be adopted, especially in the context of this paper, as different insulating phases are disregarded. Could the authors either adopt 65 C as T_c or provide reasoning regarding why they use 67 C, which is a relic from old studies on bulk or epitaxial films?

Response #1: This comment points out the controversy in T_{MIT}^0 of VO₂. We understand that T_{MIT}^0 may vary depending on the crystal form, quality, measurements, definition, etc. We believe each work should set their own T_{MIT}^0 based on their own samples and experiments, to be self-consistent. In this work, T_{MIT}^0 of VO₂ is determined by optical inspection. In Fig.

S3(b), with temperature going up, the abrupt change of optical contrast happens at 67°C, consistent with previous work on similar materials (Cao et al. Nano Lett.,10, 2667(2010)). In addition, the sharp transition indicates absence of interfacial strain between our transferred VO₂ microbeams and the substrate (Nature Nanotechnology, 4, 732 (2009)).

Comment #2: Could the authors comment on the presence of M1 and M2 phases in the insulating state of VO₂ and how this might affect their conclusions? As they relieved the nonuniform strain due to the substrate adhesion, their crystals should be in either M1 or M2 phase.

Response #2: M2 phase has been ruled out in this work because M2 phase typically exists with large external tensile stress (Park et al. Nature 500, 431 (2013), Cao et al. Nano Lett. 10, 2667 (2010)). In this work, VO₂ microbeams are transferred to and loosely sitting on the substrate without substrate adhesion or interfacial stress (Cao et al, Nature Nanotechnology, 4, 732 (2009); Wu et al, Nano Lett. 6, 2313(2006)). This has been proved by the sharp phase change as depicted in Fig. 3(b).

Comment #3: The authors assumed that hydrogen fraction in VO₂ (x) causes a linear change in the T_{MIT}^0 . This is backed up by the ab initio calculations in Ref.19 (in the manuscript) as well as from the gradual shift of the M-I domain wall gradually shift to the other tip when heated towards T_{MIT}^0 . However, although samples doped above 100 C for a short duration exhibit such behavior, samples doped at lower temperatures for longer durations show less sharp M-I boundaries, as in Ref. 6. This is also somewhat evident in Figure S6b, as from 2 to 8 to 16 hours the resistance change doesn't seem to follow a sharp boundary. Could the authors explain how this faint M-I boundary fits into the thermodynamic picture they propose?

Response #3: A faint boundary may be an interesting effect, but the boundaries in our experiments are all sharp.

(1) The reviewer may misunderstand Fig. S6(b): In Fig. S6(b), from 0 to t_1 to t_2 to t_3 ,

$L_M \sim \sqrt{Dt}$. Considering a series resistor below: R is dominated by $(L - L_M)$ because $\rho_M \ll \rho_I$. The data approximately shows how L_M varies with t , and is not telling whether the boundary is faint or not. Fig. S6(b) as well as its captions have been modified to be clearer.

(2) Absence of faint boundary: We suppose the reviewer is talking about Fig. 3a in Ref.6 for a faint boundary, while a faint boundary is not observed in our work. In our annealing experiment below T_{MIT}^0 ($T_{anneal} < T_{MIT}^0$), where the annealing temperature is even lower, and duration is even longer, the M-I boundary in samples is sharp. A few more

optical images are shown below besides the ones in Fig.2(b) and Fig. S4.

Comment #4: Although W-doped measurements somewhat serve that purpose, could the authors elaborate more on the spillover rate of the Pd catalyst at low temperatures, i.e. below T_c ? One peculiar aspect related to the spillover rate is much long durations required for hydrogen doping of W-doped VO_2 as compared to the undoped case above T_c . This brings up the issue that if the Pd spillover is slower at lower temperatures along with the lower diffusion coefficient of the insulating phases, there should be an additional correction to the calculated γ factor.

Response #4:

- (1) What the reviewer said about the spillover rate may be correct. However, if there is a factor of the spillover rate versus temperature, it should also exist in the region above $T_{\text{MIT}}^0 = 67^\circ\text{C}$ (blue line in Fig.1(c)). In other words, when only considering this factor, $\ln D$ should follow a straight line without a kink, opposite to Fig. 1(c).
- (2) As the reviewer has pointed out, the W-doped measurements are designed to exclude other factors, such as the one that the reviewer mentioned. In Fig.4, we are comparing the hydrogen diffusion rate in I- VO_2 and M- $\text{W}_{0.015}\text{V}_{0.985}\text{O}_2$ ($T_{\text{MIT}} = 30^\circ\text{C}$) at the same temperature (37°C), rather than one below T_{MIT}^0 , one above T_{MIT}^0 . Captions of Fig. 4 have been revised to avoid confusion.
- (3) To exclude influence from the dopant (W), we have performed additional diffusion experiments on VO_2 and $\text{W}_{0.015}\text{V}_{0.985}\text{O}_2$ both at 90°C , so that they are both in the M phase. In Fig. S12, it clearly shows that the diffusion rate of hydrogen in M- VO_2 and M- $\text{W}_{0.015}\text{V}_{0.985}\text{O}_2$ at the same temperature is close to each other. However, the diffusion is significantly retarded in I- VO_2 compared to M- $\text{W}_{0.015}\text{V}_{0.985}\text{O}_2$ at the same, low temperature (37°C). Therefore, it is mainly the phase being M or I that causes the diffusion retardation. Fig.S12 is also shown below.

Comment #5: Relevant to point 4, in Note S4, the authors mention how electrical measurements on tungsten-doped polycrystalline samples are used to measure the diffusion depth. However, no conductivity data is presented. It would be helpful to include the conductivity results in the supporting information.

Response #5: The conductivity results have been added as Table S5 in the supplementary information.

Comment #6: A scale bar to the OM inset of Fig2b would be helpful.

Response #6: Thanks for the suggestion to improve the quality. A scale bar as well as the corresponding caption has been added to the inlet photo in Fig. 2(b).

REVIEWERS' COMMENTS

Reviewer #1 (Remarks to the Author):

I read through the revised manuscript and also the reply to my previous comment. The author well addressed my previous concerns.

Reviewer #2 (Remarks to the Author):

The current version of draft is fine to me and I recommend to publish it the NC.

Reviewer #3 (Remarks to the Author):

I have no further comments for this paper.

Reviewer #4 (Remarks to the Author):

The authors have addressed all my concerns and made the necessary changes in the manuscript. I think it is suitable for publication in its current form.

As a note, I would like to add that although I am convinced that the type of insulating phase plays no significant role in this study, I still think my previous comments on the phase transition temperature and the possibility of having an M2 phase are valid and perhaps should be considered for future studies. If the authors could check the attached file I modified from Park et al. Nature (2013) Fig.4, the red arrows I added to the phase diagram hint that the observed 67 C phase transition might be due to slight uniform tensile stress on the crystals as a result of subsequent fabrication steps after transferring the crystals.

[**Editorial note:** Reviewer 4's modified figure is hereby redacted to remove third-party material where no permission to publish could be obtained.]